# Modelling seasonality of Lassa fever incidences and vector dynamics in Nigeria

**James Q. McKendrick**[1]*, **Warren S. D. Tennant**[2], **Michael J. Tildesley**[2]

**1** MathSys, Mathematical Institute, Zeeman Building, University of Warwick, Coventry, United Kingdom,
**2** Zeeman Institute: SBIDER, School of Life Sciences and Mathematics Institute, University of Warwick, Coventry, United Kingdom

* j.mckendrick@warwick.ac.uk

**Data Availability Statement:** NCDC situation reports for lassa fever are readily available at https://reliefweb.int/country/nga. Codebase and specific time-series data used can be found in the public github repository at https://github.com/

## Abstract

Lassa fever (Lf) is a viral haemorrhagic disease endemic to West Africa and is caused by the *Lassa mammarenavirus*. The rodent *Mastomys natalensis* serves as the primary reservoir and its ecology and behaviour have been linked to the distinct spatial and temporal patterns in the incidence of Lf. Nigeria has experienced an unprecedented epidemic that lasted from January until April of 2018, which has been followed by subsequent epidemics of Lf in the same period every year since. While previous research has modelled the case seasonality within Nigeria, this did not capture the seasonal variation in the reproduction of the zoonotic reservoir and its effect on case numbers. To this end, we introduce an approximate Bayesian computation scheme to fit our model to the case data from 2018–2020 supplied by the NCDC. In this study we used a periodically forced seasonal nonautonomous system of ordinary differential equations as a vector model to demonstrate that the population dynamics of the rodent reservoir may be responsible for the spikes in the number of observed cases in humans. The results show that in December through to March, spillover from the zoonotic reservoir drastically increases and spreads the virus to the people of Nigeria. Therefore to effectively combat Lf, attention and efforts should be concentrated during this period.

## Author summary

Lassa fever is a viral disease prevalent in West Africa, with *Mastomys natalensis* serving as the primary reservoir. In Nigeria, annual outbreaks occur from December to March. Using a novel model and data from 2018–2020, we demonstrate that the population dynamics of the reservoir contribute to spikes in human cases. Specifically, spillover transmission increases drastically during this period, highlighting the need for concentrated efforts and interventions. Understanding the seasonal dynamics of the reservoir is crucial for effective Lassa fever control and prevention strategies in Nigeria.

JimmyMckendrick1/LassaFever_
SeasonalityNigeria.

**Funding:** JM was funded by the Engineering and Physical Sciences Research Council through the MathSys CDT [grant number EP/S022244/1]. Additionally, MJT and WT were funded on a joint BBSRC/EEID grant [Grant Number BB/T004312/1]. The funders had no role in the study design, data collection and analysis, decision to publish or preparation of the manuscript.

**Competing interests:** The authors have declared that no competing interests exist.

## Introduction

Lassa fever (Lf) is a viral zoonotic disease, caused by the *Lassa mammarenavirus* (LASV), that is endemic to West African countries such as Nigeria, Sierra Leone and Guinea [1, 2]. Lf has a natural reservoir in the rat *Mastomys natalensis* in which the virus persists and crossover events to humans occur [3–5]. The disease, which was first described after two nurses contracted the disease in a hospital in Jos, Nigeria, in 1969, has since been identified as a significant risk to health in West Africa with 300, 000–500, 000 cases per year resulting in approximately 5, 000 deaths annually [3, 6, 7]. Those infected with LASV typically experience acute symptoms of headaches, sore throat, muscle pain, vomiting and diarrhoea, and in severe cases bleeding from the mouth, nose, vagina or gastrointestinal tract [1]. The risk that Lf poses to public health will only increase without widespread intervention and a viable vaccine as growth in inter-border traffic and international travel increases the likelihood of introducing the virus to other regions within and outside of the African continent [8, 9].

In recent years, Nigeria has experienced epidemics with peak incidence occurring between December and April just after the rainy season ends in October since at least 2016 with epidemics since 2018 being notably more severe [10]. Both the number of cases and the exposure rate of Lf increases in certain periods of the year and have been correlated with rainfall patterns [11–13]. This may be because the reproduction of *M. natalensis* is greatest just after the rainy season, which results in an increase in the size of the rat population and in the spread of LASV from infected to susceptible rats [14–17]. The ecological dynamics of *M. natalensis* are relevant to Lf in humans because the majority of infections (80%) are suspected to be spillover events as opposed to human-to-human transmissions [18]. The importance of rat contact with humans is illustrated by data reported by Tobin et al. (2015) for Edo state, Nigeria, which recorded 32.4%(385/1189) of the confirmed national cases of Lf in 2020; 96.1% houses had found the multimammate rat within them in the past 6 months and 58.2% of the resident were seropositive (i.e. tested positive for Lf-specific antibodies) [19]. Transmission from rats to humans can occur in a variety of ways, such as consumption of the rats, and contamination of food from urine and faeces [5, 20]. While human-to-human transmission can occur, it is through transmission of bodily fluids and occurs predominantly in health care settings in the absence of adequate infection prevention and control measures [1]. Hence the relationship between the presence and behaviour of *M. natalensis* and the prevalence of Lf is critical to understanding and predicting future outbreaks.

Mathematical modelling studies of Lf are rare compared with other diseases, despite the inclusion of Lf in the World Health Organization's Blueprint list of diseases to be prioritized for research and development [21]. Published models for recent outbreaks in Nigeria can incorporate the population dynamics of the disease reservoir and highlight areas and periods of the year at high risk of transmission [22–24]. Akhmetzhanov et al. (2019) used suspected case data and a rodent model to inform a time-dependent exposure rate of Lf to susceptible people in Nigeria, however relied on various parameter estimates for the vector model from other studies rather than fitting to the data [11]. Furthermore, many models focus on a single epidemic, potentially overlooking important nuances in transmission and vector population dynamics within the regular pattern of incidence data observed in Nigeria [25–29]. We fit to 3 consecutive epidemics so that we may elucidate transmission dynamics that might be missed in their separation. To date, the mechanistic models describing Lf epidemics in Nigeria lacked the focus on time-dependent parameters relating to the rodent population dynamics to explain the relationship between those dynamics and the seasonality of outbreaks.

In this study, we developed an epidemiological model to describe the temporal dynamics of Lf within Nigeria in both human hosts and rodent vectors incorporating seasonal variations in

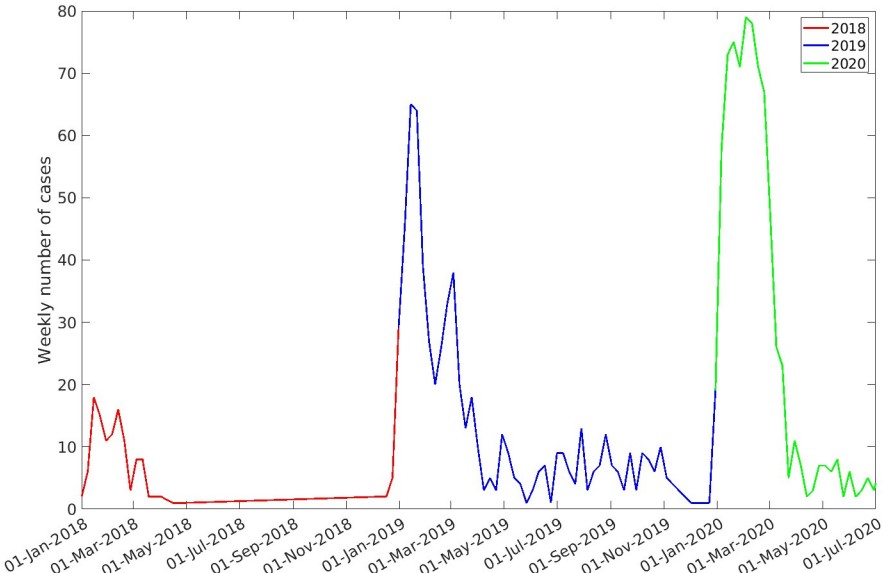

**Fig 1. Weekly confirmed case data for Lassa fever in Nigeria between the weeks ending 7th January 2018 until 12th July 2020.**

rodent population and recruitment dynamics. The model was then fitted to confirmed case data from the Nigerian Centre for Disease Control (NCDC) containing the 2018, 2019 and 2020 epidemics (see Fig 1) at the national scale in Nigeria using Approximate Bayesian Computation (ABC). This approach allowed us to demonstrate how the annual fluctuations of the rodent population translated into the seasonal outbreaks of Lf cases in human hosts.

We believe that providing an epidemiological model of the dynamics of Lf in humans and the rodent reservoir within Nigeria as a whole that fits to confirmed cases supports the hypothesis that the fluctuations of the local rodent population strongly influence the cases observed. In order to inform future public health efforts, it is important to better understand the role of the disease reservoir in the spatio-temporal profile of infections.

## Materials and methods

### Ethics statement

No primary data was collected as part of our study, and thus no ethics approval was required. Secondary data used in our study—reported Lassa fever cases across Nigeria from 2018—2020 —were collected as part of the national surveillance programme lead by the Nigerian government and the National Lassa fever Emergency Operations Centre.

### Data

The weekly situation reports on Lf produced by the NCDC provided a stream of publicly available data that were discussed in regular direct communications with NCDC representatives. The data used in this study was the set of dated Lf cases collected by the NCDC's surveillance network for Lf between 7th January 2018 until 12th July 2020, which covers the 2018, 2019 and 2020 epidemics [30]. We have chosen this period to investigate the seasonal dynamics in Lf in Nigeria since it is both an extended period covering multiple consecutive epidemics and also has comparable levels of surveillance for each epidemic. While surveillance in Nigeria has

resulted in data being available for prior years, 2018 is notable as 2 more laboratories capable of providing diagnostic services opened and increased the capacity [10]. It was observed that cases increased in previous years similar to that seen in the data presented; however, the 2018 epidemic was significantly larger than those of 2016 and 2017. Likewise, for the 2021 season we suspect that nationwide interventions during the COVID pandemic affected reporting as we see more than a 50% decrease in cases from 2020 [31]. Therefore we would likely need to account for the difference in diagnostic capacity if these years were included. Lf cases were categorised as either suspected, confirmed and probable. Confirmed cases were those which had a positive result for IgM antibody, PCR or virus isolation. Suspected cases were any individual experiencing symptoms such as fever, sore throat, vomiting, diarrhoea. Additionally they also met one of the following criteria: if they had a history of contact with either 1) excreta or urine of rodents, 2) with a probable or confirmed Lf case recently, or 3) any person with inexplicable bleeding/haemorrhagia. Due to the uncertainty that would result from use of unconfirmed data we used only confirmed cases.

## Model

In order to describe the confirmed Lf cases in Nigeria, we constructed a vector-host model in which the population and transmission dynamics of the rodent *M. natalensis* are modelled explicitly in addition to the dynamics of human transmission and disease progression (Fig 2 and Eqs 1–3). Table 1 refers to all the populations and subpopulations detailed in the model.

The human population was split into susceptible, exposed, asymptomatic or infected, and recovered ($S_h$, $E_h$, $A_h$, $I_h$, $R_h$). The infection pathway for humans was described in the model as follows: Susceptible people acquire infections from the pool of all infectious individuals. This is at a rate of $\lambda_h$. Contact with an infectious host, whether it is a human or vector rodent, transmits the disease to the susceptible individual and then they become exposed. Exposed humans become infectious after a period of $1/v$ days, and thus exposed individuals will leave the compartment at a rate of $v$. Those who were exposed to the virus will then become Asymptomatic with a probability of $p$ and Infected with a probability of $1-p$. Infected humans are assumed to experience more severe symptoms and be recorded as cases in data; they also experience an infection induced mortality rate $\mu_{h_I}$. Infected individuals are assumed to be detectable. Infected and Asymptomatic individuals will recover with an average recovery period of $1/\gamma_h$ days whereby they have protective immunity.

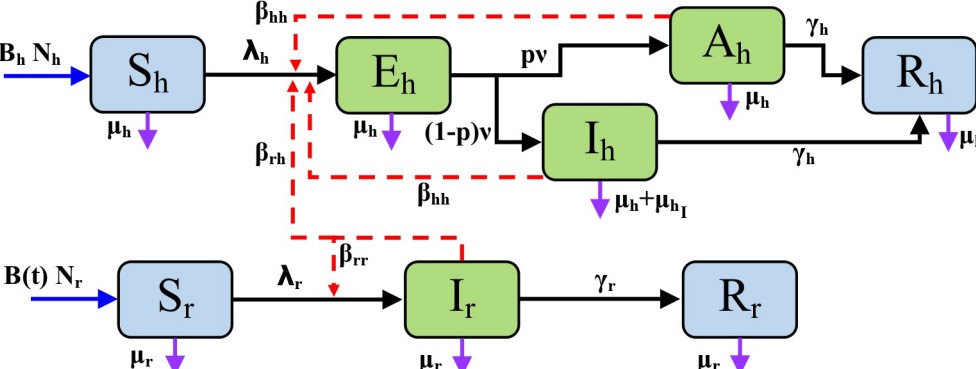

**Fig 2. Model flowchart of the transmission and population dynamics of the system of Eq 3.** Blue solid arrows denote recruitment. Black solid arrows denote progression of the disease. Red dashed arrows denote disease transmission. Purple solid arrows denote mortalities. Parameters are detailed in full in Table 2 where $\lambda_h$ and $\lambda_r$ are defined in Eq 2 (i) and (ii) respectively, and $B(t)$ is defined in Eq 1.

Table 1. Description of compartments of the model in (3).

| Compartment | Description |
|---|---|
| $N_h$ | Total population of humans |
| $S_h$ | Number of humans susceptible to Lf |
| $E_h$ | Humans that have been exposed to Lf |
| $A_h$ | Humans infected with Lf and are asymptomatic |
| $I_h$ | Humans burdened with symptomatic Lf |
| $R_h$ | Humans that have recovered from Lf |
| $N_r$ | Total population of rats |
| $S_r$ | Susceptible rats |
| $I_h$ | Infected rats |
| $R_h$ | Recovered rats |

Humans without burden of Lf are assumed to have a natural life expectancy of $1/\mu_h$ days. An infection induced mortality rate, $\mu_{h_I}$ is added to this for symptomatic individuals. Humans are assumed to have a constant birth rate, $B_h$, and newborns are assumed to be fully susceptible to Lf.

The vector population of rodents was split into susceptible, infected, and recovered ($S_r$, $I_r$, $R_r$). We note that these rodents are also in contact with humans and so must represent members of the population in close proximity to rural homes. Rats are assumed to be born susceptible and are recruited into areas in close-proximity to people at a time-dependent per capita rate of $B(t)$. The rat infections follow a more simple pathway than that found in humans: Susceptible rats come into contact with infected rats, whereby the disease is transmitted to the susceptible individual. This occurs at a rate of $\lambda_r$. The infected rats do not experience increased mortality as they are asymptomatic and thus the rats have a constant mortality rate of $\mu_r$. Susceptible rats become infectious without lag and will recover with an average recovery period of $1/\gamma_r$, where they also enjoy protective immunity.

*M. natalensis* has repeatedly been observed to have seasonal breeding habits over different areas of Africa [14–16, 32–36]. Therefore we model recruitment seasonally with a rate per capita of the rats $B(t)$ was chosen as

$$B(t) = k \exp\left\{-s \cos\left(\left(\pi\left(\frac{t}{365} - \phi\right)\right)\right)^2\right\} \tag{1}$$

where $k$ is the magnitude of the function; $s$ is a shape parameter denoting how long the period of low reproduction rates lasts for, a smaller $s$ meaning a close to constant recruitment rate over the year whereas a larger $s$ would equate to a long low period then a sharper change to a high period; and $\phi$ is the point in the year where the reproduction of the rats is at its minimum. $k$ and $s$ are of positive value, and $\phi$ is between 0 and 1. Once parameters $s$ and the natural mortality rate of the rats, $\mu_r$, have been fixed, $k$ may be scaled to keep the year-on-year reservoir population stationary [37]. A population whose yearly dynamics are similar is a better representation of a species that is endemic to the environment and although fluctuations will happen, the data to accurately model the population does not exist.

The rate at which susceptibles become infected, otherwise known as the force of infection, is denoted by $\lambda_h$ and $\lambda_r$ for humans and rats respectively. In humans, this is defined as a linear combination of the possible contact routes with LASV carriers. To represent the difference in transmissibility of Lf in rats and humans, the contact rates $\beta_{rh}(t)$, $\beta_{rr}$ and $\beta_{hh}$ are separated. We have assumed that human-to-rat transmission does not occur in the model as there have been

no studies detailing the infection path in the literature and when modelled it is often ignored or neglible in comparison to other transmission routes [11, 20, 23, 38–40]. Explicitly, these parameters are the rate of successful transmissions from rats to humans per infected rat, transmissions from rats to rats per infected rat and transmissions from humans to humans per infected human respectively. In addition, we have assumed a density dependent contact rate between susceptible humans and the pool of infectious individuals, which means that contacts will occur at an invariable rate irrespective of the size of the human population. Furthermore, observations of the vector have shown that during the dry season, higher abundance of *M. natalensis* have been trapped in or close to human dwellings, increasing the likelihood of transmission. Therefore, $\beta_{rh}(t)$ is doubled during the wet season between November and March at its peak than that during the rest of the year as observed with trapping success data [13].

With the transmission rates between compartments defined, the susceptible compartments experience a force of infection from the infectious agents. The total force of infection per susceptible is therefore the sum linear combination of the number of infectious agents that can infect that susceptible host multiplied by the appropriate transmission rate. Therefore the force of infection per individual human and rat, respectively denoted $\lambda_h$ and $\lambda_r$, are as follows:

$$\lambda_h = \frac{\beta_{rh}(t)I_r + \beta_{hh}(A_h + I_h)}{N_h}$$

$$\lambda_r = \frac{\beta_{rr}I_r}{N_r} \qquad (2)$$

$$\frac{dS_r}{dt} = B(t)N_r - \lambda_r S_r - \mu_r S_r$$

$$\frac{dI_r}{dt} = \lambda_r S_r - (\gamma_r + \mu_r)I_r$$

$$\frac{dR_r}{dt} = \gamma_r I_r - \mu_r R_r$$

$$\frac{dS_h}{dt} = B_h N_h - \lambda_h S_h - \mu_h S_h$$

$$\frac{dE_h}{dt} = \lambda_h S_h - (v + \mu_h)E_h \qquad (3)$$

$$\frac{dA_h}{dt} = pvE_h - \mu_h A_h$$

$$\frac{dI_h}{dt} = (1-p)vE_h - (\gamma_h + \mu_{h_I} + \mu_h)I_h$$

$$\frac{dR_h}{dt} = \gamma_h I_h - \mu_h R_h$$

## Basic reproduction numbers

The basic reproduction ratio, $R_0$, is the expected number of secondary infections caused by a single infectious agent in an otherwise susceptible population. Over time, the conditions which the infectious agent inhabit change and thus we also calculate the effective reproduction rate, which shows the expected number of secondary infections from one infectious agent in the population (which may have other infectious agents) at a specified time $t$.

Based on the next generation method for deriving the reproductive ratio, $R_0$, from Diekmann et al (1990) and the particular method used in van den Driessche and Watmough's work

(2002) we produce the reproductions ratios between species in equation set 4 with derivations in S1 Appendix (Section 1.3) [41, 42]. The effective reproduction rates at time $t$ are shown in 7.

$$R_0^{rr} = \frac{\beta_{rr}}{(\gamma_r + \mu_r)} \tag{4}$$

$$R_0^{rh} = \frac{\beta_{rh}}{(\gamma_r + \mu_r)} \tag{5}$$

$$R_0^{hh} = \left( \frac{pv\beta_{hh}}{(v + \mu_h)(\mu_h + \gamma_h)} + \frac{(1-p)v\beta_{hh}}{(v + \mu_h)(\mu_{h_I} + \mu_h + \gamma_h)} \right) \tag{6}$$

$$R^{rr}(t) = \frac{\beta_{rr}}{(\gamma_r + \mu_r)} \frac{S_r}{N_r} \tag{7}$$

$$R^{rh}(t) = \frac{\beta_{rh}}{(\gamma_r + \mu_r)} \frac{S_h}{N_h} \tag{8}$$

$$R^{hh}(t) = \left( \frac{pv\beta_{hh}}{(v + \mu_h)(\mu_h + \gamma_h)} + \frac{(1-p)v\beta_{hh}}{(v + \mu_h)(\mu_{h_I} + \mu_h + \gamma_h)} \right) \frac{S_h}{N_h} \tag{9}$$

## Parameter selection

In Table 2 we show the choices for the model parameters that are described in this subsection.

Outbreak events within Nigeria have a seasonal pattern, thus each epidemic should not be analysed in isolation as this would neglect the effect of seasonal dynamics. Therefore the data fitting takes place over a longer time period and the population dynamics of the people of Nigeria should be taken into account. The natural death rate of humans $\mu_h$ is estimated to be $\frac{1}{54 \times 365}$ day$^{-1}$ since 54 years was the average life expectancy to 2 significant figures for Nigerians

**Table 2. Description of parameters of the model in 3.**

| Parameter | Biological Description |
|:---:|:---:|
| $B_h$ | Human birth rate |
| $\mu_h$ | Human natural mortality rate |
| $v$ | Reciprocal of incubation period |
| $\gamma_h$ | Human recovery rate |
| $p$ | Probability of an infectious human being asymptomatic |
| $\mu_{h_I}$ | Infection induced mortality in humans |
| $\gamma_r$ | Recovery rate for rats |
| $\mu_r$ | Natural mortality rate for rats |
| $\beta_{rr}$ | Rat-to-rat transmission rate |
| $\beta_{rh}$ | Rat-to-human transmission rate |
| $\beta_{hh}$ | Human-to-human transmission rate |
| $\phi$ | Time of minimum recruitment for rats |
| $s$ | Shape parameter for recruitment function for rats |
| $k$ | Magnitude of recruitment rate for rats |

given by the World Bank for 2018. The human birth rate $B_h$ was approximated as $1.2 \times 10^{-4} \text{day}^{-1}$. Nigeria has experienced close to exponential growth rate in recent years and if it is assumed that the population growth of Nigeria has been stable over this time period then we may assume that $\frac{dN}{dt} = (B_h - \mu_h)N_h$, where $B_h$ and $\mu_h$ are constant. Therefore $N(t) = N(0)e^{(B_h - \mu_h)t}$. We then obtain $B_h = \mu_h + \frac{1}{T} \log \frac{N(T)}{N(0)}$. The growth of Nigeria from 2015 to 2019 was 181.1 million to 201.0 million to 4 significant figures which gives the growth rate of Nigeria to be $7.14 \times 10^{-5}$ per capita per day hence giving $B_h = \mu_h + 7.14 \times 10^{-5}$ [43].

The incubation period was assumed to be 14 days. There is a wide range for the incubation period and is reported to be around 2 days to 3 weeks. For simplification this assumption was made to be approximately 14 days and thus $v = 1/14$. The rate of recovery for humans $\gamma_h$ was 0.1 day$^{-1}$. Ranges for the time to recover are broad, between 2 and 21 days, so similarly to $v$ a value of 0.1 was assumed. We assume that the probability of becoming asymptomatic $p = 0.8$ [1].

The infection induced mortality rate, $\mu_{h_I}$, is derived as follows. The proportion of those that have died, retrospectively, during the outbreak period considered is 196/1006. Therefore the case fatality rate (CFR) is 19.5% to 3 significant figures [30]. Since in the model CFR $= \frac{\mu_{h_I}}{\mu_{h_I} + \gamma_h + \mu_h}$ we obtain $\mu_h = 0.0242$. While we acknowledge the potential for under-reporting of deaths due to difficulties in posthumous diagnosis, our study relied on the available data to maintain consistency.

The initial number of infected humans $I_h(0)$ is 2 since there were two recorded confirmed cases in the week commencing 01/01/2018. Therefore $A_h(0) = 8$ to maintain the ratio between asymptomatic and symptomatic infected persons to 1:4. $E_h(0)$ is 5 times that of the number of cases reported the week after the data being used starts. This is done to represent that those who show symptoms were likely exposed the week before and that 20% of the exposed will go on to show symptoms. The initial total number of people living in Nigeria was assumed to be $2 \times 10^8$. The initial number of recovered individuals $R_h(0)$ was assumed be 30% of the total population as this was within the range found in previous studies for those that had previously encountered Lf [5, 19, 44, 45]. Since estimates for serological positivity in people can vary substantially between study locations we conducted sensitivity analysis on $R_h(0)$ with 10% and 20%. The remaining population were assumed susceptible $S_h(0)$.

The mortality rate of rats is assumed as $\mu_r = 0.0038$ per day since this value has been previously used as the baseline value in previous works [46]. $\gamma_r = 1/90$ per day [47]. For the population size of the zoonotic reservoir we kept $N_r(0) = 1$. The number of initially susceptible rats was $S_r(0) = N_r(0)/R_0^{rr}$. This is necessarily bounded by $N_r(0)$ above and below by 0 since $\beta_{rr}$ may be sampled such that $R_0^{rr} < 1$. $I_r(0)$, the total initial proportion of infectious rats, is fitted to allow exploration of dynamics that may not be the model's endemic equilibrium.

The remaining parameters, $\phi$, $s$, $\beta_{rr}$, $\beta_{rh}$, $\beta_{hh}$ and $I_r(0)$ will be estimated within the fitting scheme in the following section. All parameters are listed in Table 2 with biological descriptions. For the assumed values of parameters see Table 3.

## Fitting and data

Bayesian estimation techniques involve a suite of statistical inference methods based on the idea that after specifying a prior assumption upon the parameters being investigated, the prior is updated with the introduction of more information from the observed data. Following Bayes' theorem, the posterior distribution of parameters is obtained by combining the prior beliefs (prior distribution) with the evidence of the data which usually comes in the form of the likelihood function [48].

**Table 3. Fixed and fitted parameters to be estimated in the model.** The parameters of interest were inferred using algorithm 1 in section 2.4.

| Parameter | Value or Prior | Fitting status |
|---|---|---|
| $N_h(0)$ | $2 \times 10^8$ | Fixed [43] |
| $E_h(0)$ | 30 | Fixed |
| $A_h(0)$ | 8 | Fixed |
| $I_h(0)$ | 2 | Fixed, see sensitivity analysis |
| $R_h(0)$ | $0.3 \times N_h(0)$ | Fixed, see sensitivity analysis [5, 44, 45] |
| $S_h(0)$ | $N_h(0) - E_h(0) - A_h(0) - I_h(0) - R_h(0)$ | Fixed, see sensitivity analysis |
| $N_r(0)$ | 1 | Fixed |
| $S_r(0)$ | $N_r(0)/R_0^{rr}$ | Parameter-dependent |
| $I_r(0)$ | $Uniform(0, 1)$ | Fitted |
| $B_h$ | $1.2 \times 10^{-4} \text{day}^{-1}$ | Fixed |
| $\mu_h$ | $\frac{1}{54 \times 365} \text{day}^{-1}$ | Fixed [43] |
| $\nu$ | $0.1 \text{ day}^{-1}$ | Fixed [1] |
| $\gamma_h$ | $0.1 \text{ day}^{-1}$ | Fixed [1] |
| $p$ | 0.8 | Fixed [1] |
| $\mu_{h_I}$ | $0.0242 \text{ day}^{-1}$ | Fixed |
| $\gamma_r$ | $1/90 \text{ day}^{-1}$ | Fixed [47] |
| $\mu_r$ | $0.0038 \text{ day}^{-1}$ | Fixed [46] |
| $\beta_{rr}$ | $LogNormal(-1.03, 1)$ | Fitted |
| $\beta_{rh}$ | $LogNormal(0.347, 1.5)$ | Fitted |
| $\beta_{hh}$ | $LogNormal(-2.35, 0.5)$ | Fitted |
| $\phi$ | $Uniform(0, 1)$ | Fitted |
| $s$ | $LogNormal(3, 1)$ | Fitted |

In the absence of a likelihood function, which may arise because the function is intractable or computationally expensive, Approximate Bayesian Computation (ABC) is a robust method that can be used. ABC can be summarised as a family of techniques where parameters are sampled, in various different ways that are dependent on the specific scheme, and then accepted or rejected if the simulated data given the parameters are sufficiently close to the observed data [49]. ABC schemes are becoming increasing popular due to their relative ease of use.

In this paper we use a modified Approximate Bayesian Computation Sequential Monte Carlo scheme (ABC SMC) to fit our parameters (see Algorithm 1 for psuedocode description of scheme used) since the scheme has been shown to be reliable and converges faster than some of the more primitive schemes [50]. The ABC SMC schema iterates a population of parameter particles over $T$ generations with decreasing tolerances, $\{\varepsilon_i\}_{i=1}^T$, allowed between the data, $y^*$, and the data simulated from the model with the particle $\theta$, $y_\theta$. This converges to the desired approximate posterior distribution as the distributions of the parameters are sequentially improved upon [51].

ABC SMC fits a model $M$ with unknown model parameters $\theta$ to data. The standard algorithm requires one to specify a decreasing sequence of thresholds $\varepsilon_1 \geq \varepsilon_2 \geq \cdots \geq \varepsilon_T$ for the $T$ generations. When starting, $t = 1$, parameters are sampled from prior distributions, $\pi(\theta)$. For each subsequent generation $t = 2, \cdots, T$ parameters will be sampled from a perturbation kernel, $q_t(\theta|\theta_{t-1}^{(i)})$, based on a sampled particle accepted in generation $t - 1$. The model is then simulated with the sampled parameter particle and using a chosen distance metric to compare the data and the simulated data, the parameters are accepted if the error calculated is smaller than the given tolerance for that generation, i.e. $d(y^*, y_\theta) \leq \varepsilon_t$.

When implementing the SMC algorithm, instead of manually defining the sequence $\varepsilon_1, \varepsilon_2, \cdots, \varepsilon_T$, which is often done by manually calibrating the tolerances after some initial test runs, we instead initialise our algorithm with an integer, $K$, and a proportion, $Q$. If the desired number of parameter particles is $N$ then we initialise by sampling $KN$ particles from the prior distributions and reject all the $N$ particles with the smallest errors. This serves as our first generation of sampling. For subsequent generations we have set a desired quantile, $Q$, where the particle that is the $Q^{th}$ quantile has its error between the data and model set as the tolerance for the next generation. That is, if the set of parameter particles for generation $g-1$ is $\{\theta_i^{g-1}\}_{i=1}^N$ and is ordered with respect to $d(y^*, y_\theta)$ then for generation $g$ the tolerance $\varepsilon_g = d(y^*, y_{\theta_Q^{g-1}})$. Furthermore the perturbation kernel that we use is a multivariate Gaussian with variance equal to twice that of the co-variance between the previous generation's particles:

$q_t(\theta|\theta_{t-1}^{(i)}) = \mathcal{N}(\theta|\theta_{t-1}^{(i)}, \Sigma_{t-1})$, where $i$ is sampled using the weightings generated.

**Algorithm 1: Pseudocode of modified SMC ABC**. This was used to fit the model in section 2.1. Instead of using an arbitrary sequence of tolerances, the tolerances are calculated from the errors produced in the previous generation. For the first generation, the algorithm runs a multiple, $K$, of the number of desired particles, $N$, and then accepts the best $N$.

**Input**: $N$, number of particles per generation

$K$, multiples of $N$ for the initial sampling to determine the sequence of tolerances $\varepsilon_i$

$Q$, Quantile between 0 and 1 to select the next tolerance from the distribution of tolerances from the previous generation

$\pi(\theta)$, Prior distribution for the tested variables

$q_t(\theta|\theta_{t-1}^{(i)})$, Method of perturbation to generate samples of particles for generations $t = 2, \cdots, T$

$y_0$, Data and method of determining closeness to simulated data $d(\cdot, \cdot)$

Model $M(\theta)$

**Output**: $\{\theta_i^T\}_{i=1}^N$, the accepted parameters from generation $T$;     `// Run the first generation`

**for** $i = 1$ *to KN* **do**

Generate $\theta^i$ from the prior $p(\theta)$ Generate data $y_{\theta^i}$ from the model $M(\theta_i)$

Calculate $d_i = d(y_{\theta^i}, y^*)$

**end**

Sort initial $KN$ particles by their distance entries $d_i$

Set $\{\theta_i^1\}_{i=N}^N$ to be the $N$ best of the $KN$ particles, (retroactively making $\varepsilon_1 = d_N$);     `// Run subsequent generations`

**for** $t = 2$ *to T* **do**

Calculate weights $\omega_1^{(i)} \leftarrow 1/N$ Set next tolerance with $Q$ by letting

$\varepsilon_2 = d_{floor(QN)}$ **while** $i = \leqslant N$ *do* **do**

Draw $\theta^*$ from among $\theta_{t-1}$ with probabilities $\omega_{t-1}$ Generate $\theta$ from

$q_t(\theta|\theta_{t-1}^{(i)}) = \mathcal{N}(\theta^*, \Sigma_{t-1})$ where $\Sigma_{t-1}$ is the covariance of the previous generation of particles Generate $y_\theta$ from the simulator $d(y_\theta, y^*) \leq \varepsilon_t$

$\theta_t^{(i)} \leftarrow \theta$ $\omega_t^{(i)} \leftarrow \frac{p(\theta)}{(\Sigma_{k=1}^N \omega_{t-1}^{(k)} \mathcal{N}(\theta|\theta_{t-1}^{(k)}, t-1))}$

**end**

$\varepsilon_{t+1} = d_{floor(QN)}$ where $floor(x)$ is the greatest integer less than $x$

**end**

**Prior distributions.**   For the parameters that we are unable to calculate, we fit with the scheme from section 2.4 specifically described as Algorithm 1. In this section we explain the

choices for the prior distributions from our prior knowledge on the model and disease epidemiology. All parameters and their priors are listed in Table 3.

$\phi$ is sampled from a uniform distribution between 0 and 1 because this does not give any preference to any period of the year over another. $s$ is sampled from a Log-normal distribution, Lognormal $(\mu_1, \sigma_1^2)$, where $\mu_1 = 3$ and $\sigma_1 = 1$ as this allows the parameter samples to be varied and comparable to the range used in Peel et al (2014) [37]. We sample $\beta_{rr}$ from a Lognormal distribution, $Lognormal(\mu_2, \sigma_2^2)$. Since the zoonotic system is assumed to be in endemic equilibrium, $R_0^{rr}$ is assumed to be greater than 1. We therefore set $\mu_2$ so that the mode of the distribution, $m_2 = \exp(\mu_2 - \sigma_2^2)$ where $\sigma_2 = 0.5$, would equate to $R_0^{rr} = \frac{m_2}{\mu_r + \gamma_r} = 10$, with a prior flexible enough to sample through parameter space for a variety of values. Therefore $\mu_2 = \log(10(\gamma_r + \mu_r)) + \sigma_2^2 = -1.65$ to 3 s.f. Similarly, for $\beta_{rh}$ we sample from a Lognormal prior distribution such that the mode $m_3$ gives an $R_0^{rh} = \frac{m_3}{\mu_r + \gamma_r} = 10$. Thus the prior distribution has $\mu_3 = \log(10(\gamma_r + \mu_r)) + \sigma_3^2 = 0.347$ to 3 s.f. with $\sigma_3 = 1.5$. For the initial proportion of the population of rats that were infected, $I_r(0)$, we assumed no preference and used a uniform prior between 0 and 1. Lastly, since human-to-human transmissions are unlikely outside of nosocomial settings, we set $\sigma_4 = 0.5$ and $\beta_{hh} \sim$ Lognormal $(\mu_4, \sigma_4^2)$ so that the mode $m_4$ when equated to $\beta_{hh}$ and inserted into $R_0^{hh}$ would give $R_0^{hh} = 1$. Therefore $\mu_4 = \log(\psi) + \sigma_4^2 = -2.35$ to 3 s.f. where $\sigma = 0.5$ and

$$\frac{1}{\psi} = \frac{pv}{(v + \mu_h)(\mu_h + \gamma_h)} + \frac{(1-p)v}{(v + \mu_h)(\mu_{h_I} + \mu_h + \gamma_h)} \tag{10}$$

As previously stated, the data used here are taken directly from our communications with the NCDC. The data points used are laboratory confirmed cases aggregated by week from the 12th of December 2018 until the 4th of April 2020.

## Implementation

The fitting algorithm and all models were implemented in MATLAB, with the models using the ODE45 solver for numerical integration of the ODEs.

The algorithm ran for $T = 15$ generations with each generation consisting of 2500 parameter particles. For the first generation $K = 10$ multiples of 2500 particles were ran and then the $1/K$ quantile with the least error was accepted. Subsequent tolerances for each generation were set with the quantile $Q = 1/6$, where errors from the previous generation would generate the next tolerance value.

## Results

We applied an ABC fitting scheme (Algorithm 1) to the model detailed in Materials and methods (Fig 2 and Eq 3) to the confirmed Lassa fever cases in Nigeria. Notably, our fitted model successfully replicated the observed seasonal trends in the data, as demonstrated in Fig 3. The posterior distributions of parameters and vector dynamics (Figs 4 and 5, respectively) reveal the seasonal nature of the epidemics and how the population dynamics of the primary reservoir, *M. natalensis*, affect the number of observed cases in Nigeria.

### Model evaluation

The model fit resulted in a final generation whose simulations of the number of symptom-presenting humans, $I_h$, can be seen in Fig 3. The entire range of values that $I_h$ takes in the final generation at each time point is in light orange and the median value in blue. Overlaid in black is the confirmed case data for Nigeria. This shows that the model can replicate the year-on-year trend and that the seasonal epidemics in Nigeria can be explained by the vector population dynamics.

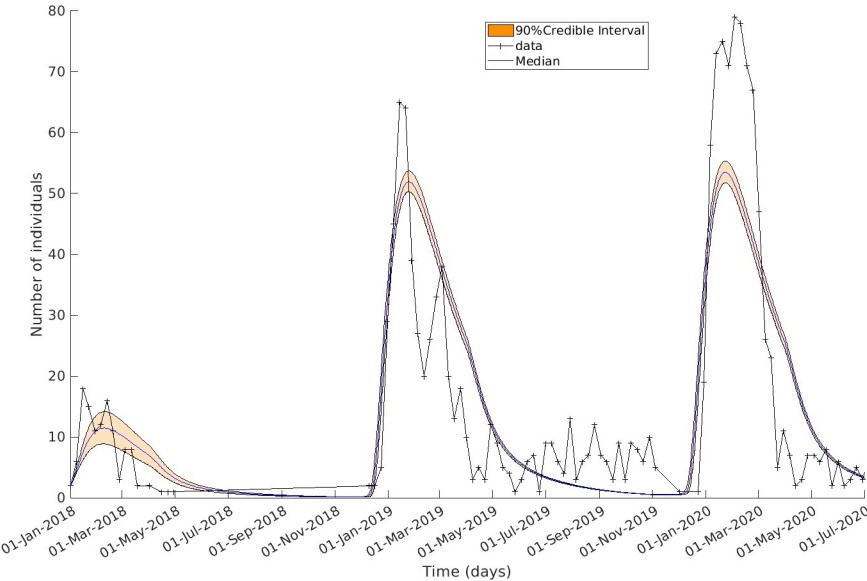

**Fig 3. The epidemiological model captured 3 consecutive Lf epidemics in Nigeria.** The simulated cases compared with the observed data. In orange is the 90% range of values $I_h$ takes in the final generation at each time point; the median value in blue. Confirmed case data for Nigeria are in black. The model replicates the sharp increase in case incidences occurring at the start of the year for 3 years.

The number of infected rats increases drastically in December when the pool of available susceptibles grows, thereby increasing the spillover rate to humans (Fig 6). As the number of rats in contact with humans decreases over the year due to natural mortality and recovery the number of spillover events decreases. This process starts again just before the next epidemic and continues cyclically.

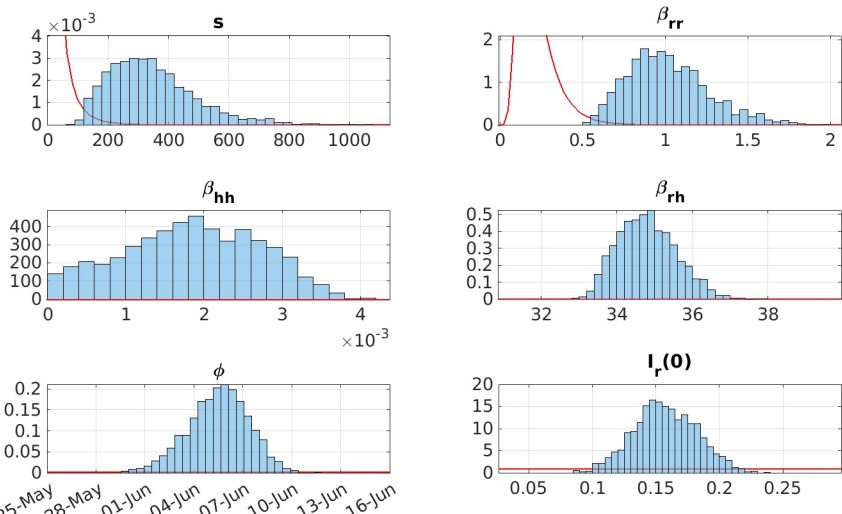

**Fig 4. The marginal posterior distributions of the final set of accepted particles from fitting.** Fig 4 top left the shape parameter of the rodent recruitment function, $s$. Fig 4 top right the rodent-to-rodent transmission rate $\beta_{rr}$. Fig 4 mid left the human-to-human transmission rate $\beta_{hh}$. Fig 4 mid right the rodent-to-human transmission rate $\beta_{rh}$. Fig 4 bottom left the date of minimum rat recruitment $\phi$.

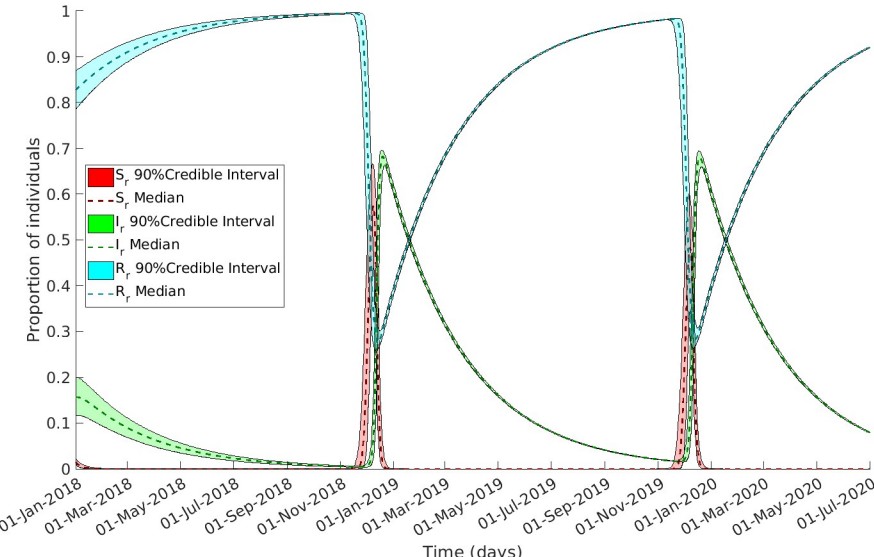

**Fig 5. Underlying vector dynamics reveal high-risk period of spill-over transmission for Nigerians.** The figure showcases the evolution of *M. natalensis* compartments throughout the observed period, simulated using the parameters derived from the final generation of accepted values. The median value is represented by the dashed line, while the colored area illustrates the range. Susceptible rats are depicted in red, infected rats in green, and recovered rats in blue. Notably, the recruitment of susceptible rats progressively rises, providing impetus for the growth of infected rats, reaching its peak in late December. Consequently, this surge in infected rats leads to spillover infections in humans.

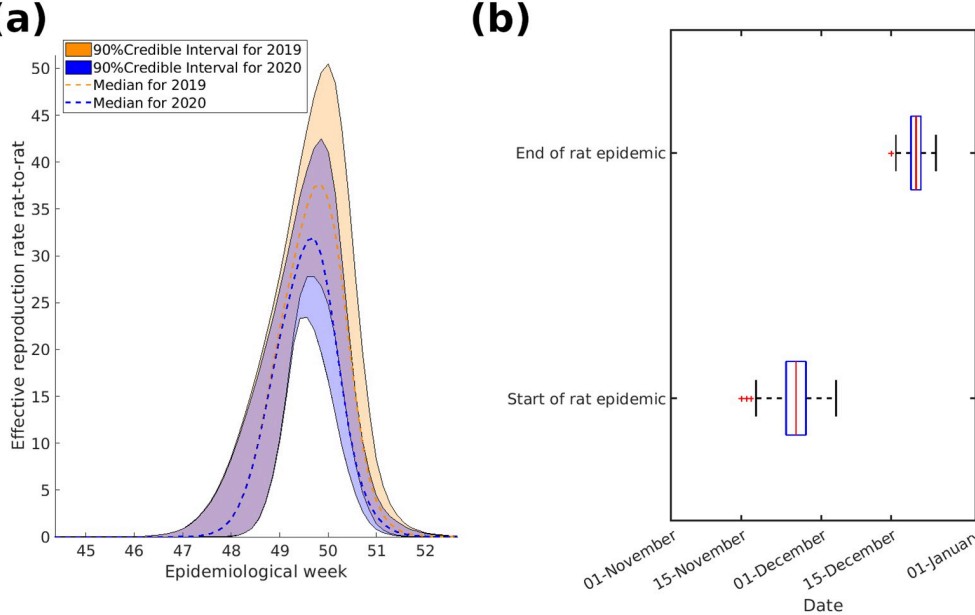

**Fig 6. The range and median of the effective reproduction rate for rat-to-rat transmission $R^{rr}(t)$ and when the threshold for $R^{rr}(t) \geq 1$ is met.** In Fig 6 (a) $R^{rr}(t)$, median dashed-line and range in coloured block, exhibits a sharp increase towards the end of the year, foreshadowing the subsequent outbreaks in the following months. To maintain clarity, the data is limited to the years 2019 and 2020, as no complete earlier records are available. Fig 6 (b) showcases a box diagram illustrating the time of year when $R^{rr}(t)$ exceeds the threshold of 1, denoting high transmission. The bottom panel of Fig 6 (b) captures the onset of the high transmission period, while the upper panel displays its conclusion. The intermediate phase witnesses a rapid shift in reservoir dynamics, leading to an escalation in the number of infected vectors.

## Marginal posterior parameter distributions

The recruitment rate of the vector *M. natalensis* is determined by the shape parameter, *s* Fig 4 top left, and the date of the minimum rate, $\phi$ Fig 4 bottom left, is focused and seasonal. Since the posterior of *s* has increased its median substantially ($3.24 \times 10^2$), this results in a large ratio between the recruitment rate's lowest and highest value and thus the rat population experiences an influx of susceptible rats at $\phi + 6$ months in early December. The distribution for $\phi$ is concentrated around a median of 0.427 (5th of June) with a 90% credible interval of 0.418–0.435 (2nd–8th June). This is observed in the rodent dynamics in Fig 5 where the number of susceptible rats increases rapidly.

The time-varying reproduction rate for rat-to-rat transmission crosses the threshold of $R_t^{rr} \geq 1$ in early December (Fig 6(a)) and causes the number of infectious rats to increase similarly. This then spills over to humans and causes the spike in incidence data that is observed in Nigeria between January and March.

The transmission parameters, Fig 4 top right and middle, are such that human infections are predominantly the result of a spillover event from the zoonotic reservoir. The proportion of humans infected by rats is estimated to be 96.2%–99.0% (90% credible interval) with a median of 97.5%.

## Biological implications

In our study, we explicitly modelled the vector dynamics allowing us to investigate biological implications. The seasonal recruitment of the natural reservoir as seen in Fig 5 is a crucial aspect of the disease dynamics and serves as a key driver of the transmission cycle. We can infer from these results that *M. natalensis* recruitment, which may be an combination of birth and migration, influence the occurrence of spillover events to humans in Nigeria, with rat-to-rat infection peaking just before the epidemics observed in Nigeria (Fig 6a and 6b). This emphasizes the importance of understanding the ecological and reproductive dynamics of the reservoir species, as it directly impacts the risk of disease transmission to human populations.

## Sensitivity analysis

We conducted a sensitivity analysis (see S1 Appendix Section 3) to assess the impact of varying the assumed proportion of people that have been exposed to Lassa fever. In the original fitting, we assumed 30% had encountered and recovered from the disease. We substituted 10% and 20% of the population into the recovered population as these values were comparable to that found in population sampling studies [5, 19, 44, 45]. When fitting under these alternate assumptions we used the unaltered SMC algorithm with the sequence of tolerances set to be those calculated in the original run as seen in Table A in S1 Appendix.

We saw a decrease in the rat-to-human and human-to-human transmission rates as the proportion of recovered individuals decreased. This correlates with the increase in proportion of susceptible people and therefore these parameters would need to decrease in order to compensate, maintaining the same epidemic sizes.

We also challenged our assumptions on the reporting rates for Lassa fever cases in 2018. As noted previously, 2018 saw the introduction of new facilities in Nigeria capable of diagnosing samples for Lf [10]. This resulted in an increase in capacity for diagnoses and likely the number of confirmed incidences observed since we are without evidence of change in the nature of Lf transmission such as genomic mutation resulting in increased infectivity [52]. However, a further jump in confirmed cases was seen later in 2019 and 2020. This may have been due to increased awareness of Lassa fever and familiarity with the health care and surveillance systems

available after the first year that the new facilities commenced operation. We therefore assumed alternative reporting rates of 33% and 50%. We conducted this analysis with the alternative assumptions for the proportion of the Nigerian population that had initially encountered and recovered from Lf.

We found that with a higher number of cases, and therefore assumed initial number of infected humans, that the transmission rate within the reservoir decreased and the parameter $\phi$ decreased as well while the initial proportion of infected rodents increased proportionately with the scaling of the 2018 data. The first epidemic in the data, that of 2018, is affected more by the initial values and so the link between the initial proportion of infected rodents and the size of the simulated data for 2018 is apparent. It appears that as the data for the 2018 epidemic reflects the later epidemics more, the model's endemic equilibrium can better represent the first epidemic. Otherwise, the fitting scheme prefers the dynamics of the zoonotic reservoir to be more focused to reach the burnout of the epidemic quicker to allow the model to capture both a small epidemic and larger epidemics later.

## Discussion

Nigeria has experienced substantial epidemics of Lassa fever in recent years. However, the drivers of these epidemics have not been explored in detail using a mechanistic model and described qualitatively. The role of the natural rodent-reservoir in the seasonality of Lf cases is therefore not well understood. To that end we fitted to the 3 consecutive epidemics in Nigeria, using data from 2018 to 2020, rather than a single outbreak in an attempt to elucidate transmission and population dynamics that might be missed in their separation. This included previously over-looked rodent population dynamics for which the possibility of seasonal recruitment was investigated. This constraint and methodology results in the dynamics shown and should initiate more investigation into the transmission dynamics of Lassa fever. We found that the model qualitatively replicated the weekly confirmed case data, showing that the seasonal peak in cases can be attributed to the population and epidemiological dynamics of the zoonotic rodent reservoir.

By fitting our model to case data from the outbreaks in Nigeria, we inferred that the recruitment rate of *M. natalensis* rats in contact with at-risk Nigerians was highly seasonal, which lead to rapid and substantial shifts in the proportion of the reservoir that were susceptible to and infected with LASV. We found that there was a pulse increase of new susceptible rats in December of each year that quickly become infected by LASV-carrying rodents surviving the previous epidemic. These infected rats then spread the infection to human inhabitants in shared rural environments. The recruitment rate for the zoonotic vector being sharp and focused in December, during the dry season for Nigeria, corroborates with previous work that showed a marked increase in *M. natalensis* breeding during the end of rainy season to up to 3 months after [14, 16, 17, 32–36, 53]. This drove the increase in observed cases in humans with over 95% transmission events being spillover events from the zoonotic reservoir. This proportion was larger than that seen in other studies, such as Iacono et al (2015), which may reflect the omission of within hospital interactions and the increased risk of transmission in hospitals without appropriate infection control precautions [18].

Our estimates for the basic reproduction number—the expected number of secondary infections from a primary infection in a fully susceptible population—of Lf in rats were well above that which one would consider to be an extremely infective disease, such as measles [54]. In vector epidemiology, however, large measurements of $R_0$ are not unusual [55, 56]. This transmissibility is coupled with a sharp increase in local rat abundance in December causing a sharp and substantial increase in infected rodents in proximity to humans, causing the

increase in the number of spillover events. It is important to note that the model's representation of rat dynamics, particularly the recruitment function, may not capture the true complexity of the population dynamics. Using a recruitment function instead of a true birth function allows for a simplified life cycle of *M. natalensis* by omitting a nesting/juvenile stage. Therefore dynamics that occur before recruitment and contact with humans may be missed, and could explain why the basic reproduction number for rats is higher than expected because potential infected individuals are excluded from transmission.

There are not many studies capturing the seasonality in the prevalence of viremia in rats. However, Fichet et al. (2007) found that it is high in the wet season and lower in the dry season [13]. This is in contrast to that observed in our model but serves to highlight what is not understood about the infection cycle of the LASV. In our model, the viremic proportion of the vector varies greatly, increasing to nearly 70% at its peak. These large proportions are comparable to localised samples in other studies such as Keenlyside et al. (1983) and Safronetz et al. (2013) [44, 57]. Furthermore, it is noted in Fichet et al. (2014) that rats in locations that are considered as high-endemic have a higher prevalence of LASV than in other areas [47]. Given that most reported cases are spillover events in high-endemic areas of Nigeria, we hypothesize that the model, which assumes homogeneous mixing on a national scale, replicates dynamics more closely resembling those of high-endemic areas.

In our sensitivity analysis we adjusted the initial number of recovered humans between 10%, 20% and 30%, consistent with proportions found in the literature. We also varied the percentage of cases that were reported in 2018, owing to its comparatively small number of cases compared to 2019 and 2020. We adjusted for 50% and 33% reporting and scaled the initial number of infected, asymptomatic and exposed humans. The model was able to capture the dynamics with the alterations to the initial proportion of infected vectors and an earlier and less severe epidemic within the rodent reservoir. We believe that pressure for the model to produce a small epidemic at the beginning of the time series but larger epidemics afterwards results in parameters that would lead to slower, protracted epidemics being rejected. Thus high transmission between rats was preferred so that the "epidemic" within the rat population reached burnout quicker, and therefore allowing the period of increased spillover risk to be both short enough for the first epidemic to not grow outside of it but sharp and rapid so that subsequent years still have sufficient exposure.

Incorporation of fine-scale space or further refinement of the rat population dynamics into the model—such as including time-dependent rat-to-rat transmission rates which may represent a hypothetical change in behaviour and proximity to humans—could improve both the model's realism and enhance our understanding of observed fluctuations in rodent seroprevalence [13]. For example, these developments may explicitly consider increased rat populations near homes in the dry season and higher LASV prevalence in *M. natalensis* during the rainy season [13, 32]. Additionally, with case data published at a higher than national scale resolution, a metapopulation model would better capture spatial variations between different administrative areas, improving the representation of disease dynamics. To enhance comprehensiveness, future studies should incorporate hospitalization and treatment options specific to human cases, offering insights into healthcare worker risks and transmission reduction strategies. These enhancements would advance understanding of the disease and improve applicability of the model, thereby helping guide better strategies for Lf control and prevention in Nigeria.

## Conclusion

Our model captured the dynamics of weekly confirmed Lf cases over multiple epidemics in Nigeria. Our approach demonstrated that the population of *M. natalensis* experienced annual

LASV outbreaks due to seasonal recruitment rates resulting in an influx of new susceptible rats to fuel the spread of the disease. It is not yet clear what proportion of this recruitment is due to seasonal reproduction or migration to rural homes in the dry season. The high number of infected vectors causes a spillover of infection, resulting in annual epidemics between late December and early April. There are only relatively low levels of human-to-human transmission, which supports the notion that the zoonotic reservoir of *M. natalensis* is the primary driver in the epidemiology of Lf. Therefore, we conjecture that the single most effective measure of controlling the epidemics would be to reduce human contact with the zoonotic reservoir, either by increased food security and hygiene, or with more effective trapping and culling of rats.

## Supporting information

**S1 Appendix. Appendix containing mathematical analysis and sensitivity analysis.**
(PDF)

## Acknowledgments

We thank the Nigeria Center for Disease Control for support and data supply.

## Author Contributions

**Conceptualization:** James Q. McKendrick, Warren S. D. Tennant, Michael J. Tildesley.

**Formal analysis:** James Q. McKendrick.

**Investigation:** James Q. McKendrick.

**Methodology:** James Q. McKendrick.

**Supervision:** Warren S. D. Tennant, Michael J. Tildesley.

**Visualization:** James Q. McKendrick.

**Writing – original draft:** James Q. McKendrick.

**Writing – review & editing:** James Q. McKendrick, Warren S. D. Tennant, Michael J. Tildesley.

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
