## [Decision Letter · Decision Letter 0]

1 Sep 2023

Dear Mr Mckendrick,

Thank you very much for submitting your manuscript "Modelling seasonality of Lassa fever incidences in Nigeria" for consideration at PLOS Neglected Tropical Diseases. As with all papers reviewed by the journal, your manuscript was reviewed by members of the editorial board and by several independent reviewers. In light of the reviews (below this email), we would like to invite the resubmission of a significantly-revised version that takes into account the reviewers' comments. 

We cannot make any decision about publication until we have seen the revised manuscript and your response to the reviewers' comments. Your revised manuscript is also likely to be sent to reviewers for further evaluation.

Sincerely,

Chukwunonso Nzelu, Ph.D.

Guest Editor

Andrea Marzi

Section Editor

Reviewer's Responses to Questions

**Key Review Criteria Required for Acceptance?**

**Methods**

-Are the objectives of the study clearly articulated with a clear testable hypothesis stated?

-Is the study design appropriate to address the stated objectives?

-Is the population clearly described and appropriate for the hypothesis being tested?

-Is the sample size sufficient to ensure adequate power to address the hypothesis being tested?

-Were correct statistical analysis used to support conclusions?

-Are there concerns about ethical or regulatory requirements being met?

Reviewer #1: (No Response)

Reviewer #2: 1. The initial number of recovered individuals is assumed to be 0 (see page 6, lines 163-4). Given that seroprevalence data are cited from 2015-era publications, it is not realistic that (nearly) the entire population would be naïve in 2018. I would expect that the model would be very sensitive to the initial number of recovered individuals. Unfortunately, no sensitivity analysis around this parameter was investigated. This is even though sensitivity analyses were performed for much less critical parameters, see comments below. Without knowing how the number of initial recovered individuals influences the model outputs, I don’t think that any useful conclusions can be drawn. It is common for epidemics in naïve populations to display “waves” of infection. Thus, how do we know that the model is reproducing the seasonal epidemics because a naïve population is being modelled, or because the model is capturing the essential nature of the disease dynamics? I would suggest either including a sensitivity analysis for the number of initial recovered individuals, or alternatively adding it to the calibration parameter set. One way or another, this manuscript should not be published without resolving this issue in a satisfactory manner.

2. I am not clear on why the authors expect the model to be sensitive to the size of the rat population. From the model equations, it seems likely that setting the rat population to an arbitrary level and then scaling the transmission coefficient \\beta_{rr} (which would happen automatically during the calibration procedure) should be sufficient for the purposes of this model. While it is useful to perform the sensitivity analysis anyways (in case of any “surprises”), I would not spend any time on this issue in the main manuscript (since no such “surprises” arise). Instead, I would strongly suggest addressing the issue discussed in point (1), which is of much more import to the overall model dynamics and the ability to draw conclusions from the results. Also, on page 6, lines 167-168 it looks like there is a typo “1, 10^6” should be “1x10^6”? N(0)=1 doesn’t seem to appear further on in the text. As another related, but minor, issue, why not take the initial values for the rat population from the stable periodic solution of the rat-specific equations (evaluated at the appropriate time)? This would simplify the initial parameter set and seems much more realistic.

3. The case fatality rate (Page 5, lines 151-156:) is equal to the flow out of the Infected class due to disease-related deaths divided by the total flow out of the compartment, i.e., CFR = \\mu_{h_I} / (\\gamma_h + \\mu_{h_I} + \\mu_h). With the parameters and CFR as reported in the manuscript this would yield \\mu_{h_I} = 0.0242 per day vs 0.0195 per day (about a 25% difference). This would probably not make a huge difference to the disease dynamics, but it will make a large difference when investigating the change in deaths due to disease after an intervention has been introduced. I would correct this issue here, rather than it become an issue in subsequent models.

Reviewer #3: The objectives of the study are clearly described and the model proposed is described in detail. The literature review is well-developed. There is a clear motivation for the statistical estimation procedure adopted.

**Results**

-Does the analysis presented match the analysis plan?

-Are the results clearly and completely presented?

-Are the figures (Tables, Images) of sufficient quality for clarity?

Reviewer #1: (No Response)

Reviewer #2: No comment.

Reviewer #3: -Does the analysis presented match the analysis plan?

Yes.

-Are the results clearly and completely presented?

Could be improved in the discussion and interpretation, but I think they are fine.

-Are the figures (Tables, Images) of sufficient quality for clarity?

- Figures could be surely improved. They do not seem of very good quality.

**Conclusions**

-Are the conclusions supported by the data presented?

-Are the limitations of analysis clearly described?

-Do the authors discuss how these data can be helpful to advance our understanding of the topic under study?

-Is public health relevance addressed?

Reviewer #1: (No Response)

Reviewer #2: The model does reproduce the observed epidemic pattern, but it is not clear how much of this is due to the assumption of a naive initial population. I don't think that the claim can be made (yet) that the model reproduces the observe epidemic behaviour because is captures the essential features of the dynamic process. In other words, I don't think that any useful conclusions can be drawn until the issues in the Methods section are resolved.

Reviewer #3: -Are the conclusions supported by the data presented?

Yes

-Are the limitations of analysis clearly described?

Could be improved

-Do the authors discuss how these data can be helpful to advance our understanding of the topic under study?

Yes

-Is public health relevance addressed?

Yes

**Editorial and Data Presentation Modifications?**

Reviewer #1: (No Response)

Reviewer #2: (No Response)

Reviewer #3: (No Response)

**Summary and General Comments**

Reviewer #1: Modelling seasonality of Lassa Fever incidences in Nigeria

The authors built and tested an ABC model to explore Lassa seasonality in Nigeria. The is the paper is very well written and easy to understand. I have specific questions below, most on the vector modeling, but my overview is that the authors argue that vector reproduction seasonality drives Lf in humans yet they admit their vector model is quite generalized. I agree that the vector model used here is somewhat primitive and the observed dynamics are unrealistic. This leads to unrealistic fitted parameters, including R0 values the authors state were 'well above that which one would consider to be an extremely infective disease' but left out of the main text (seen in table S2). A vector R0 of 135-1030 seems high, and aligns with my comments below on LASV rat transmission being too high and the population dynamics unsustainable. My suggestions for the authors are to take their own comments (lines 351-371) and improve their vector transmission model to hone in on more appropriate parameters. I was also a bit confused as they authors referenced fig5a and 5b, but there were only 4 figures in the manuscript...

line 9: LASF? Is this supposed to be Lf or LASV?

line 16: years since 2017, yet abstract says 2018? Was it introduced in 2017 and 2018 was the first 'bad' year?

line 19: M. natalensis, rather than Mastomys?

line 20: Maybe state Mastomys natalensis is a rat prior to just coming across it here?

lines 24-26: rat bites? rat feces? fomite transmission?

line 68: population dynamics of Mastomys natalensis?

line 82: Do asymptomatic individuals recover at the same rate?

line 86: Are recovered individuals (R) assumed to have protective immunity? Same for the rats?

lines 91-92: all infected rats are thus asymptomatic? 

line 100: How do you keep the reservoir population constant with seasonal breeding and constant deaths? Do you mean the avg population from year to year?

line 108: also betarr? How was betarr parameterized? What is the seasonal distribution of Ir (peak vs off season)?

line 113: Is there an assumption that no human to rat transmission occurs?

line 154: Others (Bausch D.G., et al. Lassa fever in Guinea: I. Epidemiology of human disease and clinical observations. Vector Borne Zoonotic Dis. 2001;1(4):269–281)) have found CFRs to be as high as 50%, why is the assumed CFT so low here? Is it possible a large number of unreported deaths occurred?

lines 157-159: You are also assuming 100% reporting. I imagine early reporting was likely bad, even if individuals were symptomatic. How do the parameters change if you estimate a higher number of infected individuals initially? I'm interested in the model sensitivity.

line 168: Which population size of rats is used for the model results/figures? 

line 283: The figure numbering seems off. Fig 4 is vector dynamics, fig 4b is the timeline of vector epidemics, but there are references and a caption for a figure 5?

How does human to human transmission occur?

Fig 4: It seems the infected vector has the same seasonality of LASV as humans. Isn't it more likely LASV transmission increases in rats during their breeding season (Nov-Jan?), and then there is some exposed period in rats prior to fecal shedding and the increase in cases in humans? Is there no lag?

Fig4: Why does it look like there are 0 susceptible rats for most of the year?, it's not clear to me why they peak directly before infections occur. Are you assuming 100% of rats are exposed to LASV every year and the only reason Lf drops in humans is because of vector susceptible burnout?

Fig4 & lines 351-371: How does the Ir population exceed the annual Sr population if Rr never become susceptible? Does transmission in rats happen so quickly that all newborn rats are immediately infected? Then only the very low number of infected individuals the following year seeds the entire new generation of rats? It is more likely the seasonal reproduction isn't so strong and there is a higher number of infected rats throughout the year. Even if the reproduction seasonality is this strong, it seems highly unlikely 100% of newborns are infected and the vector circulation nearly dies out every year. Whether that means a longer infected period, some sort of latent (E) period, or lowered infectiousness I don't know. The dynamics seen in Fig 4 seem unrealistic. Seasonal reproduction has variation, and you are one 'bad' or 'late' reproductive season away from eliminating LASV in the vector. I see you have addressed some of the issues with the vector population in the lines listed in the results. I think the authors need to expand and address some of the above issues. The authors need to expand because some of the fitted parameters are likely going to be inaccurate if a more accurate vector population model is used.

line 365: I'm not sure a metapopulation model would be appropriate because you have country-level reporting. If you had higher resolution data this may be possible.

conclusion: The authors conclude that seasonal vector reproduction dynamics drive Lf in humans. This comes directly after 20 lines (351-371) in the results explaining that their vector model is likely inaccurate and needs to be improved. It is already known that M. natalensis is the primary driver of Lf. 

line 375: Do the rats get Lf or LASV?

Reviewer #2: The authors present a (non-age-stratified) compartmental ordinary differential equation model for the dynamics of Lassa fever in Nigeria. The key features of the model are that it is vector-based ODE model that is primarily driven by the vector. The model successfully reproduces seasonal epidemics observed from 2018 onwards. Overall, the paper is well written and engaging and I think that the model has potential, however, there is one critical issue that needs to be addressed before publication, see Methods.

Reviewer #3: The paper is interesting and well-written. The methodology adopted seems adequate and developed with care. 

The only concern that I have about the model proposed is that it does not seem to capture the increasing level of seasonality through the years. The seasonal behavior in 2018 is quite low, starting to increase evidently in 2019 and even more in 2020. Apparently, there is an evolution in the seasonal behavior, which the model seems not able to capture. Is this a neglectable aspect of the process or should we care about it? Should we expect that in the years 2021, 2022, and 2023 this seasonal behavior will be even stronger? 

By the way, why did the authors use just the data from 2018 to 2020? 

I understand that there could be problems with the availability of more recent information, but this should be better explained in the data description. 

Also, I would suggest the authors display the data at some point at the beginning of the paper, because when I was reading I was expecting to see a graphical display of the problem. Seeing it just in the results section is probably too late.

PLOS authors have the option to publish the peer review history of their article (what does this mean?). If published, this will include your full peer review and any attached files.

Reviewer #1: Yes: Kevin Bakker

Reviewer #2: No

Reviewer #3: No
---

## [Decision Letter · Decision Letter 1]

30 Oct 2023

Dear Mckendrick,

We are pleased to inform you that your manuscript 'Modelling seasonality of Lassa fever incidences and vector dynamics in Nigeria' has been provisionally accepted for publication in PLOS Neglected Tropical Diseases.

Best regards,

Chukwunonso Nzelu, Ph.D.

Guest Editor

Andrea Marzi

Section Editor

Reviewer's Responses to Questions

**Key Review Criteria Required for Acceptance?**

**Methods**

-Are the objectives of the study clearly articulated with a clear testable hypothesis stated?

-Is the study design appropriate to address the stated objectives?

-Is the population clearly described and appropriate for the hypothesis being tested?

-Is the sample size sufficient to ensure adequate power to address the hypothesis being tested?

-Were correct statistical analysis used to support conclusions?

-Are there concerns about ethical or regulatory requirements being met?

Reviewer #1: (No Response)

Reviewer #2: (No Response)

**Results**

-Does the analysis presented match the analysis plan?

-Are the results clearly and completely presented?

-Are the figures (Tables, Images) of sufficient quality for clarity?

Reviewer #1: (No Response)

Reviewer #2: (No Response)

**Conclusions**

-Are the conclusions supported by the data presented?

-Are the limitations of analysis clearly described?

-Do the authors discuss how these data can be helpful to advance our understanding of the topic under study?

-Is public health relevance addressed?

Reviewer #1: (No Response)

Reviewer #2: (No Response)

**Editorial and Data Presentation Modifications?**

Reviewer #1: (No Response)

Reviewer #2: (No Response)

**Summary and General Comments**

Reviewer #1: The authors addressed all my main concerns and I believe it is a well-written and informational manuscript in its current form.

Reviewer #2: The authors have adequately addressed my concerns. Congratulations on a very nice manuscript.

PLOS authors have the option to publish the peer review history of their article (what does this mean?). If published, this will include your full peer review and any attached files.

Reviewer #1: No

Reviewer #2: No

---

## [Editor Report · Acceptance letter]

7 Nov 2023

Dear Mr MCKENDRICK,

We are delighted to inform you that your manuscript, "Modelling seasonality of Lassa fever incidences and vector dynamics in Nigeria," has been formally accepted for publication in PLOS Neglected Tropical Diseases.

Best regards,

Shaden Kamhawi

co-Editor-in-Chief

Paul Brindley

co-Editor-in-Chief
